# Clinical outcomes and patient-reported measures in HCV care: Insight from a longitudinal prospective study in a large Italian region

Giaele Moretti[1], Maria Paola Tramonti Fantozzi[2], Ilaria Corazza[1], Erica De Vita[2]*, Milena Vainieri[1], Lara Tavoschi[2], on behalf of OPT-HepaC project team¶

**1** Management and Healthcare Laboratory, Institute of Management and Department EMbeDS, Scuola Superiore Sant'Anna, Pisa, Italy, **2** Department of Translational Research and New Technologies in Medicine and Surgery, University of Pisa, Pisa, Italy

¶ Membership of the OPT-HepaC project team is listed in the Acknowledgments.
* erica.devita@unipi.it

## Abstract

### Background

Hepatitis C virus (HCV) infection remains a critical public health issue worldwide. Direct-acting antivirals (DAAs) have revolutionized the treatment of hepatitis C. However, real-world elimination efforts are hindered by barriers in diagnosis, treatment access, and follow-up. Embedding patient-reported outcome measures (PROMs) and patient-reported experience measures (PREMs) into routine care may improve service delivery. This study evaluates clinical and patient-reported outcomes in the HCV care cascade in Tuscany (Italy), offering insights into how health service organization affects effectiveness, equity, and patient experience.

### Methods

We conducted a multicenter, longitudinal, prospective study on 953 adults with HCV chronic infection. These adults were treated between 2021 and 2023 in seven prescribing centers in Tuscany. Clinical data included demographics, comorbidities, fibrosis staging, virological response (SVR12), and loss to follow-up (LTFU). PROMs and PREMs were collected at baseline (T0), 3 months post-treatment (T1), and 6 months after T1 (T2). We used the SF-12 tool and custom surveys. Patients were stratified by referral source (GPs, harm reduction services/prison, specialists). Clinical and questionnaire data were analyzed separately. Statistical analyses included ANOVA, Chi-square, Kruskal-Wallis, Cochran's Q, Friedman, and repeated measures ANOVA with Bonferroni corrections. Significance was set at $p \leq 0.05$.

**Data availability statement:** The data underlying the findings of this study contain sensitive information and cannot be made publicly available due to legal and ethical restrictions related to the protection of personal data and patient confidentiality. These restrictions are imposed by the Comitato Etico Regionale per la Sperimentazione Clinica della Regione Toscana – Sezione Area Vasta Nord Ovest, which approved the study protocol. Access to the minimal data set required to replicate the findings of this study may be granted upon reasonable request to qualified researchers, subject to approval by the Ethics Committee and in accordance with applicable data protection regulations. Requests for data access should be addressed to the Ethics Committee at: comitato.etico-ceavno@ao-pisa.toscana.it.

**Funding:** This work is part of the OPT-HepaC project funded by the Tuscany Regional Health Service (Regione Toscana), 2018 Health Research Tender (grant number I58D20000530002 to L.T.). The funder had no role in study design, data collection and analysis, decision to publish, or preparation of the manuscript.

**Competing interests:** NO authors have competing interests.

## Results

DAA therapy achieved high clinical efficacy: SVR12 was reached in 93.4% of patients. The rate rose to 98.6% when excluding those lost to follow-up. Patients referred by harm reduction/prison services were younger and mostly male. They had higher psychiatric comorbidities and risk behaviors. PROMs demonstrated significant improvements in perceived physical and emotional health following treatment, particularly among individuals referred by GPs and specialists. PREMs revealed increasing satisfaction with referring doctors over time. Satisfaction with specialist care remained high and stable. Referral pathways markedly influenced patient profiles and reported outcomes. There were notable disparities in experience and quality-of-life indicators.

## Conclusions

This study highlights the importance of integrating clinical and patient-reported data in monitoring HCV care. High SVR rates confirm the effectiveness of DAAs, while PROMs and PREMs provide valuable insight into patient engagement and equity of access. Stratified analyses reveal the need for tailored approaches across care pathways, and high-risk populations require special attention. Embedding patient voice in evaluation fosters a more responsive, people-centered health system, advancing progress toward HCV elimination.

## Introduction

Hepatitis C virus (HCV) infection represents a serious global public health issue, leading to severe liver disease [1,2]. The introduction of direct-acting antivirals (DAA), which are extremely effective and well-tolerated by the HCV patients [3], offered a potential pathway to reach the World Health Organisation (WHO)'s goal to eradicate HCV infection by 2030 [4]. Since 2014 in Italy, DAA treatment has been available free-of-charge, but only for patients with severe forms of HCV-related disease. Its distribution has been closely monitored by the Italian Medicines Agency (AIFA). In 2017, AIFA expanded access to include all HCV patients, regardless of disease stage [5]. However, beginning in 2019, the number of individuals treated each year began to decrease (https://www.aifa.gov.it/en/aggiornamento-epatite-c), while an estimated 398,610 HCV-infected individuals in Italy remain undiagnosed [6], effectively constituting a hidden reservoir of infection. Furthermore, some patients who do receive a diagnosis fail to initiate therapy, or are lost to follow-up, adding to the challenge of sustaining the pace needed to reach the WHO elimination targets [4].

In line with the WHO objectives of eliminating viral hepatitis, Italy has implemented a free nationwide screening program aimed at early HCV detection and treatment to prevent adverse clinical outcomes [7]. Beginning in 2021, this national health policy targets key populations, specifically, persons who use drugs (PWUDs) covered by harm reduction services (SerD), people living in prison (PLP), and general population

cohorts born in 1969–1989 [8]. Given that healthcare in Italy is organised at the regional level, each region is responsible for translating national directives into a locally tailored approach. In this framework, the Tuscany Region has created an action plan designed to enhance HCV treatment uptake and contribute to overall disease eradication by prioritising screening among PLP, PWUD, and individuals born between 1969 and 1989 [9,10].

Individuals infected with HCV experience significant vulnerabilities and are particularly affected by social and structural determinants of health [11]. These determinants include factors such as income levels and access to education [12], availability of healthcare services [13], and living conditions, such as overcrowded housing [14]. Together, they often reflect various forms of discrimination and contribute to marginalisation or social exclusion [13,15]. A patient-centred approach is fundamental for advancing care quality, as it aligns with patients' inherent right to receive dignified, respectful, and effective care [16,17]. Moreover, prioritising personalised care enhances healthcare utilisation, which plays a pivotal role in improving outcomes and contributing to the elimination of HCV [18]. Tools useful to measure and improve the quality of care provided by healthcare systems are the patient-reported outcome measures (PROMs) and patient-reported experience measures (PREMs) questionnaires, which assess the care quality monitoring clinical measures such as effectiveness, safety and patient experience [19]. Indeed, while PROMs are standardised questionnaires that acquired patients' perceptions of their health status related/not related to specific disease (levels of impairment, disability, health-related quality of life), PREMs assess patients' experiences while receiving care. Even though patient experience and satisfaction measures are now recognised as indicators of process and outcome, few examples exist in the literature on systematically collecting these measures for patients with HCV [20], compared to slightly more extensive research on patient-reported outcome measures among individuals living with chronic HCV infection [20–28]. Systematic monitoring of patient-reported measures is particularly important given this under-representation, which may be partly due to the vulnerability and difficulty in reaching populations such as people who inject drugs, those in prison, or the elderly. Despite PREMs/PROMs underutilization [29], they render sociopolitical processes more visible, potentially leading to an improvement of patient care by identifying problems and priorities [30,31]. PREMs have also been integrated into performance evaluation systems, allowing for the assessment of healthcare quality by directly incorporating patients' perspectives on their care experiences [32,33]. Indeed, improvement in health care delivery requires a deliberate focus on quality of health services, which involves providing effective, safe, people-centred care that is timely, equitable, integrated and efficient [34].

To assess the progress against the HCV elimination goals in the Tuscany region in Italy, we characterised HCV patients treated by the Tuscany region in the period 2021–2023, considering the clinical and the PREMs/PROMs responses. Notably, a study protocol has already been published describing the longitudinal collection and analysis of PREMs and PROMs in patients living with HCV in Tuscany [35]. Following the abovementioned protocol, we focused on reported outcomes, experiences, and specific characteristics of individuals who arrived at an HCV diagnosis and were linked to care through different path: general practitioners (GPs), harm reduction services/prison healthcare services, or specialists.

## Materials and methods

### Subjects

According to the intention-to-treat (ITT) analysis, this multicenter retrospective study included 953 adult HCV patients who started the HCV treatment between January 1st, 2021, and December 31st, 2023. These patients were referred to 7 out of 10 prescribing centres within the Regional Health Services of Tuscany (Italy): three teaching hospitals with infectious diseases and/or gastroenterology departments and four secondary hospitals with only infectious diseases departments.

### Sociodemographic and clinical measures

For each patient, data on baseline characteristics were gathered, including demographics (age, gender, nationality), information on medical history (HIV/HBV coinfections) as well as on HCV genotype, baseline HCV RNA level, fibrosis stages

(F0: lack of fibrosis, F1: minimal fibrosis, F2: moderate fibrosis, F3: severe fibrosis, F4: advanced fibrosis/cirrhosis) and HCV treatment with its starting date. The degree of liver fibrosis was determined by the clinicians through non-invasive methods (i.e., elastometry) and/or indirect serum markers like the FIB-4 and APRI scores. All data were directly uploaded by the clinicians on an electronic form developed by the Regional Health Agency (ARS) or on hospital databases in an anonymised way, after the approval by the patients signing a written informed consent.

Nationality was classified as Italian or non-Italian and all the other data (except for age, liver fibrosis, baseline HCV RNA value and genotype) were classified using yes/no responses.

The patient's record also documented the virological response to HCV therapy using real-time Polymerase Chain Reaction (PCR) 12 weeks after the end of treatment (EOT). Data were categorised into different groups for patients achieving or not achieving sustained virologic response (SVR12/non-SVR12) and for those who were lost to follow-up during/after DAA treatment.

## Questionnaires

Questionnaires were administered during the initial clinical visit, useful for the HCV treatment prescription (T0), during the first (T1: three months after the EOT) and the second (T2: six months after T1) follow-up visits. An in-depth description of the questionnaire is reported in the previously published study protocol [35]. The patient, after having signed an informed consent and under his/her preference, filled out the questionnaires in digital or paper-based formats. The number of patients who filled the questionnaires was 347, 237, and 104 at T0, T1 and T2, respectively. Table 1 shows the content of the surveys delivered at T0, T1, and T2. Specifically, differently from PREMs/PROMs questions present also in T1 and T2, the T0 pre-treatment questionnaire acquired information on socio-demographic characteristics and HCV history (risk behaviours and co-morbidities) of the patients and their lived experience, providing an overview of the path followed by them to identify possible best practices related to the organisational models. While the T0 PREMs/PROMs questionnaire acquired baseline data, the T1 and T2 PREMs/PROMs questionnaires were mainly useful to evaluate any improvement or worsening of health status, of well-being, as well as of patient-reported experience measurements following the HCV treatment with respect to T0.

**Table 1. Questionnaires were delivered to HCV patients at three different time points. T0: pre-treatment. T1: three months after the EOT. T2: six months after T1. (A) Dimensions investigated. (B) Number of questionnaires collected.**

| Time Points | A. Dimensions investigated | B. Questionnaires collected (n) |
|---|---|---|
| T0 | Socio-demographic information | 347 |
| | Patient's history of disease | |
| | Information on HCV diagnosis | |
| | Information on the visit with the specialist | |
| | Patient-provider relationship | |
| | Patient-reported outcomes | |
| T1 | Patient-provider relationship | 237 |
| | Overall experience along the care pathway | |
| | Patient-reported outcomes | |
| T2 | Patient-provider relationship | 104 |
| | Overall experience along the care pathway | |
| | Patient's preferences | |
| | Patient-reported outcomes | |

PROMs were collected using a scale derived from the 'Medical Outcomes Study Short- Form 12- item survey instrument' (MOS SF- 12). The MOS SF- 12 includes the following domains: physical functioning; role of physical disability; body aches and pains; general health; vitality; social functioning; role of emotional disability; and mental health. The SF-12 allows for the calculation of physical component summary (PCS) and mental component summary (MCS) scores. Scores range from 0 to 100, with higher values denoting a better quality of life. These data were collected by the Management and Healthcare (MES) Laboratory – Scuola Superiore Sant'Anna (Pisa, Italy). Pseudonymized data were accessed for research purposes on October 1$^{st}$, 2024. This study was conducted in accordance with the Declaration of Helsinki and all applicable national regulations. The research protocol was reviewed and approved by the local ethical committee (Comitato Etico Regionale per la Sperimentazione Clinica della Regione Toscana – Sezione Area Vasta Nord Ovest) on December 17, 2020 (approval number: 18829). All participants provided written informed consent prior to inclusion in the study. The study was observational in nature and classified as non-profit. Data were collected and handled in compliance with relevant data protection and privacy regulations, and analyses were conducted on anonymized datasets.

## Statistical analysis

Although we know which patients who started therapy filled out the T0 questionnaire, we were unable to link the clinical and the questionnaire data. For this reason, we separately analysed these two datasets. Statistical analysis was conducted using SPSS (Statistical Package for Social Sciences, version 20) and the significance level was set at $p <= 0.05$.

- *Whole population: general characteristics, baseline clinical assessment and clinical outcome*

Data from both hospital databases and the ARS electronic platform were combined for analysis. For the whole population of HCV patients starting the HCV treatment between 01/2021-12/2023, as well as for the related patients who did and did not fill the questionnaire at T0, descriptive statistic (expressed as mean ± standard deviation (SD), median and interquartile range, frequency (%) or number (n)) was used to describe their general characteristics, their baseline clinical assessment and their clinical outcome. To compare these variables between patients who did and did not fill the questionnaire at T0, we used one-way analysis of variance (ANOVA) for continuous variables and the Chi-square Test of Independence for binary or multi-categorical variables. In addition to the one-way ANOVA test, possible differences in age between the two groups of patients were also evaluated using the median test.

- *Questionnaire analysis*

Descriptive statistic was used to describe the answers collected by the questionnaires in the whole population, as well as in patients who have reached the HCV diagnosis through in-depth analysis prescribed by general (n = 216), specialist (n = 66) or harm reduction services/prison (n = 55) physicians. To compare the three different groups, the Kruskal-Wallis test and the one-way ANOVA were used for categorical/Likert scale and continuous data, respectively. In cases of statistically significant results, Bonferroni's post-hoc test was applied for multiple comparisons. Furthermore, for the entire population and each group, a temporal analysis of PREMs/PROMs (T0/T1/T2) was conducted. Cochran's Q test and Friedman test were utilised for binary and non-binary responses, respectively. When the results of Cochran's Q and Friedman tests were statistically significant, post-hoc Dunn's and Wilcoxon tests were respectively performed to identify specific pairwise differences.

To evaluate differences in PSC12 and MSC12 at T0 among the three groups, one way ANOVA was performed for each parameter. Moreover, a 3 Time (T0, T1, T2) repeated measures ANOVA was performed for each parameter in the whole sample, as well as in each group analysed. When necessary, adjustments for non-sphericity were applied. Post-hoc comparisons were conducted using Bonferroni correction.

## Results

### HCV patients: General characteristics

This multicenter retrospective study involved 953 HCV patients, who started the relative treatment during the period 01/2021-12/2023. Overall, in this population, the mean age was 56.31 ± 15.57 years, there were more males (61.9%) and 87.3% of the individuals were Italian. Genotype 3 (28.0%), followed by genotype 1a (25.6%) and genotype 1b (22.8%), was found to be the most common one. The liver fibrosis was lack or mild (F0/F1) in 40.9%; moderate (F2) in 19.6%, severe (F3) in 10.9% and cirrhosis (F4) was found in 28.6%. The baseline HCV RNA level was 5.91 ± 1.03 $\log_{10}$IU/ml. Coinfections with HIV and HBV were found in 1.7% and 1.8% of patients, respectively. Overall, HCV patients were mainly treated with glecaprevir/pibrentasvir (50%) and sofosbuvir/velpatasvir (47.9%), followed by elbasvir/grazoprevir (1.3%) and sofosbuvir/velpatasvir/voxilaprevir (0.8%). Twelve weeks post-treatment, HCV RNA was undetectable (SVR12) in 93.4% of patients. The remaining population was composed of patients with detectable HCV RNA (non-SVR12, 1.3%) or lost to follow-up (LTFU, 5.3%). After the exclusion of HCV patients who were LTFU, SVR12 was achieved in 98.6%. These descriptive statistics are summarised in Table 2, Column 2A. No significant difference ($\chi2(3)$=1.646, p = 0.649) was observed in liver fibrosis between HCV patients who, twelve weeks after the end of treatment, achieved a sustained virologic response (SVR12) and those who did not (non-SVR12).

*Differences between patients who filled and did not fill the T0 questionnaire* – Significant differences were observed between patients who filled (n = 347, Table 2, Column 2B) and not filled (n = 606, Table 2, Column 2C) the T0 questionnaire. In particular, the population of patients who did not fill the T0 questionnaire was younger and with a higher LTFU rate than the other one.

### Questionnaires in the whole population

From the questionnaire analyses, we removed the cases occurring when a respondent did not know how to answer the question or preferred not to answer, filling in the relative options.

*T0 questionnaire: Sociodemographic Characteristics and Clinical History of HCV Patients* – HCV patients who filled the T0 questionnaire were mainly employees (35.6%) or retired (34.0%) individuals with a middle school education (44.8%), mainly Italian (88.9%) and residing in central regions (72.9%). The most common reported risk factors were unprotected sex (68.7%), injection drug use (34.5%), and unsafe tattoo/piercing (23.7%). Other risk factors included blood transfusions/dialysis/organ transplants before 1990 (18.3%) and sharing contaminated materials (17.6%). The intrafamilial transmission by direct blood contact from the mother to child was rare (2.8%). Additionally, 17.5% had a history of detention, 2.6% were current detainees, and 8.8% were blood donors before 1990. Regarding co-morbidities, 48.5% of HCV patients had other conditions, primarily cardiovascular diseases (36.0%) and diabetes (22.0%). Other co-morbidities included oncologic (13.4%), neurological (9.8%), autoimmune (9.8%), lung (8.5%), endocrine (7.9%), and psychiatric (5.5%) diseases. The prevalence of HBV and HIV reported was 3.7% and 3.0%, respectively. These data are illustrated in Supplementary Table 1 in S1 File.

*T0 questionnaire: Referring Doctor – Diagnosis and Behaviour* – Mainly, HCV discovery occurred more than 12 months before T0 (61.7%), through routine blood tests (84.6%). The time between in-depth HCV analysis and result delivery often exceeded a week (57.5%). Additionally, most patients (61.7%) were referred directly to a specialist by their primary care physician. Overall, accessing primary care was considered easy by 91.9% of the population. These data are illustrated in Supplementary Table 2 in S1 File.

*T0 questionnaire: Care Pathway in Specialist Medicine* – Hepatologists (56.7%) and infectious disease specialists (38.9%) were the main specialists in the management of these HCV patients, seeing them within one week (38.3%) from the first call and in the easiest location to reach (94.9%). During the first visit, specialists often viewed the medical analysis brought by the patient (76.9%), visited the patient (51.9%) and performed exams such as echography (37.2%) and elastography (14.7%). A second appointment

**Table 2. HCV patients' characteristics.** Descriptive statistics of the sociodemographic characteristics, baseline clinical assessment and outcomes. Data refer to the whole population, as well as to patients who filled out (A) and did not fill (B) the T0 questionnaire. Comparisons between these latter populations are highlighted in the last column (A vs B).

| | | 2A | 2B | 2C | |
| --- | --- | --- | --- | --- | --- |
| | | Whole Population (2021–2023) (n = 953) | Population which filled the T0 questionnaire (n = 347) | Population which did not fill the T0 questionnaire (n = 606) | 2B vs 2C |
| Age | *Mean±SD (years)* | 56.31 ± 15.57 | 58.21 ± 15.95 | 55.22 ± 15.25 | **F(1,951)=8.215 p=0.004** |
| | *Median (years)* | 56.00 (46.00-66.50) | 59.9 (47.00-68.00) | 55.0 (46.00-64.00) | **χ2(1)=16.097 p<0.0005** |
| | *n* | 953 | 347 | 606 | |
| Gender | *Males (%)* | 61.9 | 59.9 | 63.0 | χ2(1)=0.896 p=0.344 |
| | *Females (%)* | 38.1 | 40.1 | 37.0 | |
| | *n* | 953 | 347 | 606 | |
| Nationality | *Italian (%)* | 87.3 | 89.9 | 85.8 | χ2(1)=3.331 p=0.068 |
| | *Not Italian (%)* | 12.7 | 10.1 | 14.2 | |
| | *n* | 951 | 346 | 605 | |
| Baseline HCV RNA level | *Mean±SD (Log₁₀ UI/ml)* | 5.91 ± 1.03 | 6.05 ± 0.94 | 5.83 ± 1.07 | F(1,924)=0.000 p=0.984 |
| | *n* | 926 | 333 | 593 | |
| Liver Fibrosis – Stages | *F0/F1 (%)* | 40.9 | 38.6 | 42.2 | |
| | *F2 (%)* | 19.6 | 20.6 | 19.0 | |
| | *F3 (%)* | 10.9 | 10.3 | 11.2 | |
| | *F4 (%)* | 28.6 | 30.4 | 27.6 | |
| | *Overall Analysis* | | | | χ2(3)=1.725 p=0.631 |
| | *n* | 919 | 339 | 580 | |
| HIV | *HIV-positive (%)* | 1.7 | 1.2 | 2.0 | χ2(1)=0.907 p=0.341 |
| | *HIV-negative (%)* | 98.3 | 98.8 | 98.0 | |
| | *n* | 949 | 345 | 604 | |
| HBV | *HBV-positive (%)* | 1.8 | 1.7 | 1.9 | χ2(1)=0.024 p=0.878 |
| | *HBV-negative (%)* | 98.2 | 98.3 | 98.1 | |
| | *n* | 925 | 343 | 582 | |
| Genotype | *1a (%)* | 25.6 | 22.7 | 27.1 | |
| | *1b (%)* | 22.8 | 25.9 | 21.1 | |
| | *3 (%)* | 28.0 | 27.4 | 28.3 | |
| | *Other (%)* | 23.7 | 24.0 | 23.5 | |
| | *Overall Analysis* | | | | χ2(3)=3.648 p=0.302 |
| | *n* | 908 | 321 | 587 | |

*(Continued)*

**Table 2.** (Continued)

|  |  | 2A | 2B | 2C |  |
|  |  | Whole Population (2021–2023) (n = 953) | Population which filled the T0 questionnaire (n = 347) | Population which did not fill the T0 questionnaire (n = 606) | 2B vs 2C |
|---|---|---|---|---|---|
| Treatment | *Glecaprevir/ Pibrentasvir (%)* | 50.0 | 45.7 | 52.5 |  |
|  | *Sofosbuvir/ Velpatasvir (%)* | 47.9 | 52.6 | 45.2 |  |
|  | *Elbasvir/ Grazoprevir (%)* | 1.3 | 0.6 | 1.7 |  |
|  | *Sofosbuvir/ Velpatasvir/ Voxilaprevir (%)* | 0.8 | 1.2 | 0.7 |  |
|  | *Overall Analysis* |  |  |  | $\chi2(3)=7.205$ p = 0.066 |
|  | *n* | 952 | 346 | 606 |  |
| Outcomes (SVR12/non-SVR12) | *SVR-12 (%)* | 98.6 | 98.7 | 98.6 | $\chi2(1)=0.044$ p = 0.833 |
|  | *Non-SVR12 (%)* | 1.4 | 1.3 | 1.4 |  |
|  | *n* | 801 | 316 | 485 |  |
| Outcomes (LTFU) | *Yes (%)* | 5.3 | 1.3 | 7.8 | **$\chi2(1)=16.921$ p < 0.0005** |
|  | *No (%)* | 94.7 | 98.7 | 92.2 |  |
|  | *n* | 846 | 320 | 526 |  |

was scheduled, and medical prescriptions were provided in 91.8% of cases. Data are illustrated in Supplementary Table 2 in S1 File. Moreover, access to specialist care was rated highly (mean scores: 4.55±0.91 on a 5-point Likert scale).

*PREMs: Referring/Specialist Doctor – Patient Relationship* – Supplementary Table 3 in S1 File shows the summary statistics of PREMs at T0 for referring and specialist doctors. Relative to the referring doctor, a significant improvement was observed in all PREMs from T0 to T1 (Table 3). These improvements were generally maintained at T2, except for feelings of being attentively cared for by medical doctors, which showed a slight decrease (Table 3). Differently, no significant temporal differences were observed in the PREMs relative to the specialist doctor (Table 4).

*PROMs* - Supplementary Table 4 in S1 File shows the descriptive statistics of PROMs at T0. HCV patients showed a significant improvement in perceived health status from T0 to T1 (p = 0.001), which remained stable over time (Table 5). Other aspects, related to physical limitations and psychological well-being, also improved significantly from T1 to T2. Since some improvements were gradual, significant differences were observed between T0 and T2: limitations in work/daily concentration and performance due to emotional state or health status, respectively (Table 5).

*PREMs: Care Pathway* - PREMs data on the care pathway followed by the HCV patients were collected in T1 (Supplementary Table 5 in S1 File) and T2. No significant differences were found between these two time points for any of the aspects analysed (Table 6).

*Preferences* – The topics of preference investigated by the survey at T2 were related to HCV rapid antibody tests information delivery, test location, post-diagnosis and treatment preferences. The results are illustrated in Fig 1.

## HCV patients with different access to the treatment path

*Sociodemographic Characteristics and Clinical History* – The analysis of sociodemographic and clinical characteristics revealed notable differences among the three groups of HCV patients, categorised based on the type of service which

**Table 3. PREMs: Patients' feedback on referring doctors at T0, T1 and T2.** Data refer to the whole population, as well as to relative groups arriving at HCV diagnosis by general, harm reduction services/prison or specialist physicians. For each aspect analysed, PREMs were based on 5-point scales, ranging from 1 (Strongly Disagree) to 5 (Strongly Agree). Multiple comparisons were performed when the Friedman test was statistically significant, by Wilcoxon tests (significance level: p<=0.017).

| | Group | n | T0 (Mean±SD) | T0 vs T1 (p) | T1 (Mean±SD) | T1 vs T2 (p) | T2 (Mean±SD) | T0 vs T2 (p) | Friedman Test |
|---|---|---|---|---|---|---|---|---|---|
| Feeling Followed in the Process | Whole Population | 81 | 4.11±1.40 | **0.007** | 4.58±0.97 | 0.431 | 4.49±1.05 | 0.023 | **χ2(2)=9.967 p=0.007** |
| | General Practitioner | 56 | 4.21±1.26 | 0.057 | 4.61±0.97 | 0.746 | 4.55±1.06 | 0.070 | **χ2(2)=6.944 p=0.031** |
| | Harm reduction services/Prison Doctor | 8 | 4.13±1.25 | | 4.38±0.92 | | 4.13±0.99 | | χ2(2)=0.615 p=0.735 |
| | Opportunistic/Specialist Doctor | 14 | 3.79±1.85 | | 4.50±1.16 | | 4.36±1.15 | | χ2(2)=3.125 p=0.210 |
| Clear Explanation of the Disease | Whole Population | 84 | 4.05±1.50 | **0.001** | 4.63±0.98 | 0.654 | 4.60±0.92 | **0.002** | **χ2(2)=10.683 p=0.005** |
| | General Practitioner | 58 | 4.09±1.45 | **0.003** | 4.71±0.88 | 0.528 | 4.64±0.93 | **0.012** | **χ2(2)=10.884 p=0.004** |
| | Harm reduction services/Prison Doctor | 9 | 4.67±0.71 | | 4.22±1.39 | | 4.56±0.53 | | χ2(2)=1.059 p=0.589 |
| | Opportunistic/Specialist Doctor | 14 | 3.86±1.75 | | 4.50±1.16 | | 4.36±1.15 | | χ2(2)=1.882 p=0.390 |
| Availability | Whole Population | 82 | 4.29±1.21 | **0.013** | 4.66±0.91 | 0.785 | 4.65±0.85 | **0.013** | **χ2(2)=10.400 p=0.006** |
| | General Practitioner | 56 | 4.43±1.01 | 0.078 | 4.70±0.89 | 0.974 | 4.71±0.80 | 0.071 | **χ2(2)=6.902 p=0.032** |
| | Harm reduction services/Prison Doctor | 8 | 4.50±0.93 | | 4.63±0.74 | | 4.50±0.76 | | χ2(2)=0.429 p=0.807 |
| | Opportunistic/Specialist Doctor | 15 | 3.80±1.70 | | 4.47±1.13 | | 4.40±1.12 | | χ2(2)=2.667 p=0.264 |
| Help to Face Fears | Whole Population | 76 | 3.97±1.45 | **0.004** | 4.51±1.04 | 0.950 | 4.49±1.00 | **0.004** | **χ2(2)=9.674 p=0.008** |
| | General Practitioner | 54 | 4.06±1.31 | **0.015** | 4.57±0.94 | 0.670 | 4.59±0.88 | **0.013** | **χ2(2)=9.188 p=0.010** |
| | Harm reduction services/Prison Doctor | 7 | 4.00±1.73 | | 4.14±1.57 | | 3.86±1.46 | | χ2(2)=1.4 p=0.497 |
| | Opportunistic/Specialist Doctor | 12 | 3.67±1.83 | | 4.33±1.23 | | 4.25±1.22 | | χ2(2)=0.7 p=0.705 |
| Feeling Involved in the Choices of the Treatment Path | Whole Population | 76 | 4.05±1.46 | **0.006** | 4.62±0.94 | 0.572 | 4.57±0.90 | **0.005** | **χ2(2)=6.271 p=0.043** |
| | General Practitioner | 54 | 4.15±1.38 | 0.025 | 4.67±0.89 | 1.000 | 4.67±0.85 | 0.020 | **χ2(2)=6.135 p=0.047** |
| | Harm reduction services/Prison Doctor | 7 | 4.43±1.13 | | 4.43±1.13 | | 4.29±0.76 | | χ2(2)=0.545 p=0.761 |
| | Opportunistic/Specialist Doctor | 12 | 3.67±1.83 | | 4.42±1.16 | | 4.25±1.22 | | χ2(2)=0.25 p=0.882 |

facilitated their access to the treatment pathway, as detailed in the method section (Supplementary Table 1 in S1 File). Patients identified by harm reduction services/prison doctors were notably younger (mean age: 45.89±11.19 years), with a higher prevalence of males (90.9%), unemployed (47.1%) and more likely to engage in high-risk behaviours, such as

**Table 4. PREMs: Patients' feedback on specialist doctors at T0, T1 and T2. Data refer to the whole population, as well as to relative groups arriving at HCV diagnosis through general, harm reduction services/prison or specialist physicians. For each aspect analysed, PREMs were based on 5-point scales, ranging from 1 (Strongly Disagree) to 5 (Strongly Agree). For each item, Friedman test was used to investigate the time effects.**

| | Group | n | T0 (Mean±SD) | T1 (Mean±SD) | T2 (Mean±SD) | Friedman Test |
|---|---|---|---|---|---|---|
| Attentively care in the Treatment Path | Whole Population | 80 | 4.77±0.50 | 4.80±0.56 | 4.78±0.60 | χ2(2)=1.156 p=0.561 |
| | General Practitioner | 53 | 4.79±0.45 | 4.85±0.53 | 4.79±0.63 | χ2(2)=2.169 p=0.338 |
| | Harm reduction services/ Prison Doctor | 8 | 4.50±0.93 | 4.38±0.92 | 4.63±0.52 | χ2(2)=0.500 p=0.779 |
| | Opportunistic/Specialist Doctor | 15 | 4.87±0.35 | 4.80±0.41 | 4.73±0.59 | χ2(2)=0.500 p=0.779 |
| Disease: Clear and Complete Information | Whole Population | 82 | 4.77±0.59 | 4.85±0.39 | 4.73±6.87 | χ2(2)=1.989 p=0.610 |
| | General Practitioner | 54 | 4.74±0.65 | 4.89±0.32 | 4.72±0.76 | χ2(2)=1.690 p=0.430 |
| | Harm reduction services/ Prison Doctor | 8 | 4.75±0.71 | 4.63±0.74 | 4.63±0.52 | χ2(2)=0.429 p=0.807 |
| | Opportunistic/Specialist Doctor | 16 | 4.88±0.34 | 4.81±0.40 | 4.75±0.58 | χ2(2)=0.500 p=0.779 |
| Therapy: Clear and Complete Information | Whole Population | 77 | 4.81±0.56 | 4.86±0.39 | 4.75±0.65 | χ2(2)=0.694 p=0.707 |
| | General Practitioner | 49 | 4.76±0.66 | 4.92±0.28 | 4.78±0.69 | χ2(2)=2.261 p=0.323 |
| | Harm reduction services/ Prison Doctor | 8 | 4.88±0.35 | 4.50±0.76 | 4.50±0.76 | χ2(2)=1.714 p=0.424 |
| | Opportunistic/Specialist Doctor | 16 | 4.88±0.34 | 4.81±0.40 | 4.75±0.58 | χ2(2)=0.500 p=0.779 |
| Therapy Duration: Clear and Complete Information | Whole Population | 79 | 4.85±0.53 | 4.85±4.26 | 4.72±0.72 | χ2(2)=2.658 p=0.265 |
| | General Practitioner | 52 | 4.83±0.62 | 4.90±0.36 | 4.73±0.77 | χ2(2)=2.213 p=0.331 |
| | Harm reduction services/ Prison Doctor | 8 | 4.88±0.35 | 4.50±0.76 | 4.50±0.76 | χ2(2)=1.714 p=0.424 |
| | Opportunistic/Specialist Doctor | 16 | 4.88±0.34 | 4.81±0.40 | 4.75±0.58 | χ2(2)=0.500 p=0.779 |
| Adverse Effects: Clear and Complete Information | Whole Population | 77 | 4.66±0.87 | 4.81±0.46 | 4.73±0.62 | χ2(2)=0.989 p=0.610 |
| | General Practitioner | 50 | 4.64±0.90 | 4.84±0.42 | 4.72±0.67 | χ2(2)=2.310 p=0.315 |
| | Harm reduction services/ Prison Doctor | 8 | 4.88±0.35 | 4.50±0.76 | 4.63±0.52 | χ2(2)=2.000 p=0.368 |
| | Opportunistic/Specialist Doctor | 15 | 4.6±1.06 | 4.80±0.41 | 4.73±0.59 | χ2(2)=0.154 p=0.926 |
| Attended Result of the Therapy: Clear and Complete Information | Whole Population | 75 | 4.72±0.67 | 4.85±0.43 | 4.72±0.71 | χ2(2)=2.366 p=0.306 |
| | General Practitioner | 47 | 4.66±0.76 | 4.94±0.25 | 4.72±0.77 | χ2(2)=5.547 p=0.062 |
| | Harm reduction services/ Prison Doctor | 8 | 4.75±0.71 | 4.38±0.92 | 4.50±0.76 | χ2(2)=1.000 p=0.607 |
| | Opportunistic/Specialist Doctor | 16 | 4.88±0.34 | 4.81±0.40 | 4.75±0.58 | χ2(2)=0.500 p=0.779 |

*(Continued)*

**Table 4.** (Continued)

|  | Group | n | T0 (Mean±SD) | T1 (Mean±SD) | T2 (Mean±SD) | Friedman Test |
|---|---|---|---|---|---|---|
| Availability for Doubts and Questions | Whole Population | 77 | 4.81±0.49 | 4.79±0.66 | 4.81±0.59 | $\chi 2(2)=0.194$ $p=0.907$ |
|  | General Practitioner | 49 | 4.82±0.49 | 4.88±0.48 | 4.82±0.63 | $\chi 2(2)=1.000$ $p=0.607$ |
|  | Harm reduction services/Prison Doctor | 8 | 4.63±0.74 | 4.13±1.46 | 4.75±0.46 | $\chi 2(2)=0.933$ $p=0.627$ |
|  | Opportunistic/Specialist Doctor | 16 | 4.88±0.34 | 4.81±0.40 | 4.75±0.58 | $\chi 2(2)=0.500$ $p=0.779$ |
| Help with Fears and Anxieties | Whole Population | 75 | 4.65±0.85 | 4.73±0.72 | 4.67±0.79 | $\chi 2(2)=1.083$ $p=0.582$ |
|  | General Practitioner | 50 | 4.64±0.83 | 4.82±0.56 | 4.74±0.69 | $\chi 2(2)=3.640$ $p=0.162$ |
|  | Harm reduction services/Prison Doctor | 7 | 4.43±1.51 | 3.86±1.57 | 3.86±1.46 | $\chi 2(2)=3.800$ $p=0.150$ |
|  | Opportunistic/Specialist Doctor | 14 | 4.71±0.61 | 4.79±0.43 | 4.71±0.61 | $\chi 2(2)=0.000$ $p=1.000$ |
| Involvement in the Choices of the Treatment Path | Whole Population | 74 | 4.72±0.79 | 4.74±0.66 | 4.59±0.89 | $\chi 2(2)=2.765$ $p=0.251$ |
|  | General Practitioner | 48 | 4.71±0.77 | 4.88±0.39 | 4.69±0.80 | $\chi 2(2)=1.870$ $p=0.393$ |
|  | Harm reduction services/Prison Doctor | 8 | 4.50±1.41 | 3.88±1.46 | 3.88±1.55 | $\chi 2(2)=3.857$ $p=0.145$ |
|  | Opportunistic/Specialist Doctor | 14 | 4.86±0.36 | 4.71±0.47 | 4.64±0.63 | $\chi 2(2)=1.600$ $p=0.449$ |

unprotected sex (96.1%) and injection drug use (90.6%). Conversely, they had a lower prevalence of historical risk factors like blood transfusions, haemodialysis, or transplants before the 1990s (5.6%). While the harm reduction services/prison doctor group had a lower overall comorbidity, psychiatric disorders were significantly more prevalent, affecting 33.3% of patients. Additionally, considering individuals currently or previously incarcerated, Kruskal-Wallis analysis ($\chi^2$ [2]=35.432, p<0.005) revealed a significantly higher prevalence of incarceration in the harm reduction services/prison doctor group (47.3%) compared to both the general (12.7%, p<0.0005) and the specialist (13.8%, p<0.0005) doctor groups. Additionally, the harm reduction services/prison doctor group had a lower proportion of retired patients (9.8%) compared to the GPs group (40.8%, p<0.0005) and a higher prevalence of patients with tattoos/piercings done in an unsafe environment (37.5%) than the specialist doctor group (15.9%, p=0.022). Related to autoimmune diseases, the GPs group (7.4%) shows a lower prevalence than the specialist one (21.6%, p=0.039).

Supplementary Table 2 in S1 File depicts the significant differences existing across the three groups for the referring and specialist care pathways.

*Referring Doctor – Diagnosis and Behaviour* – The elevated prevalence of injection drug use among patients in the harm reduction services/prison doctor group was reflected in their distinct pattern of HCV diagnosis. A significant proportion (31.3%) were diagnosed through routine blood tests conducted as part of drug addiction treatment, significantly higher than the other groups (both: p<0.0005).

Additionally, significant differences in HCV diagnosis and behaviour were observed between the specialist doctor group and the other two groups. The specialist group had a significantly lower proportion of patients who were diagnosed with HCV 12 months before T0 (38.5%; harm reduction services/prison doctor: 77.8%, specialist doctor: 65.4%, both: p<0.0005) and who were

**Table 5. PROMs: Patients' feedback on specialist doctors at T0, T1 and T2.** Data refer to the whole population, as well as to relative groups arriving at HCV diagnosis through general, harm reduction services/prison or specialist physicians. For each aspect analysed, PROMs were based on Likert scales (from 1 to 6 or from 1 to 5 or from 1 to 3) or binary responses (Yes/No). In the Likert scales, HCV patients expressed the level of agreement, where the minimum value represents the most disadvantageous feedback related to the relative outcome perceived. For binary and not binary responses, the Cochran's Q and the Friedman tests were conducted, respectively. Post-hoc Dunn and Wilcoxon tests were performed when the Cochran's Q test and the Friedman test were statistically significant, respectively. The significance level was set at p<=0.017.

| | Group | Scale | n | T0 (Mean±SD) | T0 vs T1 (p) | T1 (Mean±SD) | T1 vs T2 (p) | T2 (Mean±SD) | T0 vs T2 (p) | Friedman Test |
|---|---|---|---|---|---|---|---|---|---|---|
| Health Status | Whole Population | 1→5 | 92 | 2.74±0.92 | **0.001** | 3.11±1.01 | 0.103 | 3.29±1.00 | **<0.0005** | **χ2(2)=18.924 p<0.0005** |
| | General Practitioner | | 63 | 2.79±0.86 | **0.003** | 3.21±1.00 | 0.339 | 3.33±1.11 | **0.001** | **χ2(2)=11.803 p=0.003** |
| | Harm reduction services/Prison Doctor | | 9 | 2.78±1.39 | | 3.22±1.30 | | 3.44±0.53 | | χ2(2)=2.240 p=0.326 |
| | Opportunistic/Specialist Doctor | | 18 | 2.56±0.92 | | 2.78±0.88 | | 3.06±0.80 | | χ2(2)=4.545 p=0.103 |
| Limitations to Moderate Physical Activities | Whole Population | 1→3 | 90 | 2.40±0.79 | 0.410 | 2.46±0.69 | **0.011** | 2.67±0.60 | **0.004** | **χ2(2)=12.219 p=0.002** |
| | General Practitioner | | 60 | 2.43±0.77 | | 2.57±0.59 | | 2.62±0.64 | | χ2(2)=4.200 p=0.122 |
| | Harm reduction services/Prison Doctor | | 9 | 2.22±0.97 | 0.180 | 1.89±0.93 | 0.038 | 2.67±0.50 | 0.102 | **χ2(2)=7.600 p=0.022** |
| | Opportunistic/Specialist Doctor | | 18 | 2.39±0.85 | 0.739 | 2.44±0.70 | 0.020 | 2.83±0.51 | 0.046 | **χ2(2)=6.500 p=0.039** |
| Limitations in Climbing Stairs | Whole Population | 1→3 | 91 | 2.38±0.77 | 0.231 | 2.47±0.75 | **0.015** | 2.69±0.65 | **0.001** | **χ2(2)=16.533 p<0.0005** |
| | General Practitioner | | 61 | 2.41±0.74 | 0.023 | 2.59±0.64 | 0.429 | 2.67±0.65 | **0.015** | **χ2(2)=10.414 p=0.005** |
| | Harm reduction services/Prison Doctor | | 9 | 2.33±0.87 | | 2.00±0.87 | | 2.67±0.71 | | χ2(2)=4.000 p=0.135 |
| | Opportunistic/Specialist Doctor | | 18 | 2.33±0.84 | 0.414 | 2.44±0.86 | 0.059 | 2.83±0.51 | 0.030 | **χ2(2)=6.222 p=0.045** |
| Last 4 Weeks: Work/Daily Lower Performance due to Health Status | Whole Population | Yes (%) | 90 | 28.9 | 0.201 | 22.2 | 0.602 | 15.6 | **0.011** | χ2(2)=6.545 **p=0.038** |
| | | No (%) | | 71.1 | | 77.8 | | 84.4 | | |
| | General Practitioner | Yes (%) | 59 | 27.1 | | 15.3 | | 15.3 | | χ2(2)=4.667 p=0.097 |
| | | No (%) | | 72.9 | | 84.7 | | 84.7 | | |
| | Harm reduction services/Prison Doctor | Yes (%) | 9 | 33.3 | | 44.4 | | 22.2 | | χ2(2)=3.000 p=0.223 |
| | | No (%) | | 66.7 | | 55.6 | | 77.8 | | |
| | Opportunistic/Specialist Doctor | Yes (%) | 18 | 33.3 | | 27.8 | | 11.1 | | χ2(2)=2.889 p=0.236 |
| | | No (%) | | 66.7 | | 72.2 | | 88.9 | | |
| Last 4 Weeks: Work or Daily Limitations | Whole Population | Yes (%) | 88 | 27.3 | | 22.7 | | 17.0 | | χ2(2)=3.812 p=0.149 |
| | | No (%) | | 72.7 | | 77.3 | | 83.0 | | |
| | General Practitioner | Yes (%) | 58 | 25.9 | | 15.5 | | 19.0 | | χ2(2)=2.947 p=0.229 |
| | | No (%) | | 74.1 | | 84.5 | | 81.0 | | |
| | Harm reduction services/Prison Doctor | Yes (%) | 9 | 33.3 | | 44.4 | | 22.2 | | χ2(2)=3.000 p=0.223 |
| | | No (%) | | 66.7 | | 55.6 | | 77.8 | | |
| | Opportunistic/Specialist Doctor | Yes (%) | 18 | 33.3 | | 33.3 | | 11.1 | | χ2(2)=3.200 p=0.202 |
| | | No (%) | | 66.7 | | 66.7 | | 88.9 | | |

*(Continued)*

**Table 5.** (Continued)

| | Group | Scale | n | T0 (Mean±SD) | T0 vs T1 (p) | T1 (Mean±SD) | T1 vs T2 (p) | T2 (Mean±SD) | T0 vs T2 (p) | Friedman Test |
|---|---|---|---|---|---|---|---|---|---|---|
| Last 4 Weeks: Work/Daily Lower Performance due to Emotional State | Whole Population | Yes (%) | 88 | 28.4 | 0.509 | 25.0 | 0.048 | 14.8 | **0.008** | **χ2(2)=7.548 p=0.023** |
| | | No (%) | | 71.6 | | 75.0 | | 85.2 | | |
| | General Practitioner | Yes (%) | 58 | 25.9 | | 24.1 | | 17.2 | | χ2(2)=1.909 p=0.385 |
| | | No (%) | | 74.1 | | 75.9 | | 82.8 | | |
| | Harm reduction services/Prison Doctor | Yes (%) | 9 | 33.3 | | 44.4 | | 22.2 | | χ2(2)=3.000 p=0.223 |
| | | No (%) | | 66.7 | | 55.6 | | 77.8 | | |
| | Opportunistic/Specialist Doctor | Yes (%) | 18 | 38.9 | 0.046 | 16.7 | 0.317 | 5.6 | **0.003** | **χ2(2)=9.333 p=0.009** |
| | | No (%) | | 61.1 | | 83.3 | | 94.4 | | |
| Last 4 Weeks: Work/Daily Decreased Concentration due to Emotional State | Whole Population | Yes (%) | 88 | 27.3 | 0.379 | 22.7 | 0.078 | 13.6 | **0.008** | **χ2(2)=7.226 p=0.027** |
| | | No (%) | | 72.7 | | 77.3 | | 86.4 | | |
| | General Practitioner | Yes (%) | 57 | 21.1 | | 21.1 | | 17.5 | | χ2(2)=0.444 p=0.801 |
| | | No (%) | | 78.9 | | 78.9 | | 82.5 | | |
| | Harm reduction services/Prison Doctor | Yes (%) | 9 | 33.3 | | 33.3 | | 22.2 | | χ2(2)=1.000 p=0.607 |
| | | No (%) | | 66.7 | | 66.7 | | 77.8 | | |
| | Opportunistic/Specialist Doctor | Yes (%) | 18 | 44.4 | 0.041 | 16.7 | 0.221 | 0.0 | **0.001** | **χ2(2)=10.889 p=0.004** |
| | | No (%) | | 55.6 | | 83.3 | | 100.0 | | |
| Last 4 Weeks: Work/Daily Activities Limitations Due to Pain | Whole Population | 1→5 | 88 | 4.18±1.120 | | 4.33±0.968 | | 4.36±0.925 | | χ2(2)=0.868 p=0.648 |
| | General Practitioner | | 58 | 4.22±1.08 | | 4.48±0.86 | | 4.29±0.97 | | χ2(2)=1.480 p=0.477 |
| | Harm reduction services/Prison Doctor | | 8 | 4.38±1.41 | | 3.63±1.41 | | 4.38±0.92 | | χ2(2)=2.800 p=0.247 |
| | Opportunistic/Specialist Doctor | | 18 | 4.00±1.14 | | 4.11±0.96 | | 4.61±0.61 | | χ2(2)=3.697 p=0.157 |
| Last 4 Weeks: How Often Calm/Serene | Whole Population | 1→6 | 89 | 4.13±1.39 | 0.398 | 4.24±1.36 | **0.014** | 4.65±1.18 | **0.002** | **χ2(2)=11.036 p=0.004** |
| | General Practitioner | | 61 | 4.23±1.43 | | 4.33±1.41 | | 4.67±1.25 | | χ2(2)=2.863 p=0.239 |
| | Harm reduction services/Prison Doctor | | 6 | 4.00±1.26 | 0.276 | 3.50±1.52 | 0.066 | 4.67±1.37 | 0.046 | **χ2(2)=6.125 p=0.047** |
| | Opportunistic/Specialist Doctor | | 18 | 3.78±1.40 | 0.188 | 4.28±1.07 | 0.399 | 4.56±0.92 | 0.116 | **χ2(2)=6.408 p=0.041** |
| Last 4 Weeks: How Often Full of Energy | Whole Population | 1→6 | 90 | 3.80±1.47 | 0.110 | 4.08±1.37 | **0.002** | 4.52±1.22 | **<0.0005** | **χ2(2)=16.444 p<0.0005** |
| | General Practitioner | | 61 | 3.87±1.48 | 0.187 | 4.16±1.42 | 0.022 | 4.54±1.22 | **0.001** | **χ2(2)=8.211 p=0.016** |
| | Harm reduction services/Prison Doctor | | 7 | 4.00±1.73 | | 3.71±1.38 | | 4.71±1.25 | | χ2(2)=2.667 p=0.264 |
| | Opportunistic/Specialist Doctor | | 18 | 3.44±1.34 | 0.117 | 3.94±1.16 | 0.253 | 4.39±1.20 | 0.057 | **χ2(2)=6.391 p=0.041** |
| Last 4 Weeks: How Long Discouraged/Sad | Whole Population | 1→6 | 91 | 4.51±1.14 | | 4.71±1.05 | | 4.76±1.06 | | χ2(2)=4.856 p=0.088 |
| | General Practitioner | | 62 | 4.66±1.02 | | 4.73±1.13 | | 4.76±1.14 | | χ2(2)=0.735 p=0.692 |
| | Harm reduction services/Prison Doctor | | 7 | 4.57±1.27 | | 4.43±1.4 | | 4.57±1.51 | | χ2(2)=0.900 p=0.638 |
| | Opportunistic/Specialist Doctor | | 18 | 3.89±1.37 | **0.013** | 4.83±0.62 | 1.000 | 4.83±0.62 | 0.024 | **χ2(2)=9.644 p=0.008** |

*(Continued)*

**Table 5.** (Continued)

| | Group | Scale | n | T0 (Mean±SD) | T0 vs T1 (p) | T1 (Mean±SD) | T1 vs T2 (p) | T2 (Mean±SD) | T0 vs T2 (p) | Friedman Test |
|---|---|---|---|---|---|---|---|---|---|---|
| Last 4 Weeks: Limits on Social Activities | Whole Population | | 87 | 3.80±1.03 | | 3.99±1.05 | | 3.90±0.98 | | χ2(2)=2.136 p=0.344 |
| | General Practitioner | | 59 | 3.85±0.94 | | 3.95±1.14 | | 3.85±1.06 | | χ2(2)=0.896 p=0.639 |
| | Harm reduction services/Prison Doctor | | 6 | 3.67±1.37 | | 4.00±1.10 | | 4.17±0.75 | | χ2(2)=1.500 p=0.472 |
| | Opportunistic/Specialist Doctor | | 18 | 3.67±1.28 | | 4.06±0.80 | | 3.89±0.76 | | χ2(2)=2.000 p=0.368 |

**Table 6. PREMs: overall experience along the care pathway, T1 and T2. Data refer to the whole population, as well as to relative groups arriving at HCV diagnosis through general, harm reduction services/prison or specialist physicians. For each aspect analysed, PREMs were based on Likert scales (from 1 to 5) and significant differences were investigated by Wilcoxon test.**

| | Group | n | T1 (Mean±SD) | T2 (Mean±SD) | Wilcoxon test (p) |
|---|---|---|---|---|---|
| Support Along the Path | Whole Population | 96 | 4.73±0.72 | 4.73±0.66 | 0.911 |
| | General Practitioner | 64 | 4.78±0.65 | 4.73±0.70 | 0.685 |
| | Harm reduction services/Prison Doctor | 10 | 4.30±1.34 | 4.70±0.67 | 0.414 |
| | Opportunistic/Specialist Doctor | 18 | 4.72±0.46 | 4.72±0.57 | 1.000 |
| Know who to contact in case of need | Whole Population | 91 | 4.70±0.75 | 4.75±0.76 | 0.649 |
| | General Practitioner | 60 | 4.73±0.71 | 4.75±0.70 | 0.774 |
| | Harm reduction services/Prison Doctor | 9 | 4.67±0.71 | 4.67±0.71 | 1.000 |
| | Opportunistic/Specialist Doctor | 18 | 4.78±0.43 | 4.72±0.57 | 0.739 |
| Recommend Primary Care Services | Whole Population | 89 | 4.75±0.66 | 4.72±0.69 | 0.752 |
| | General Practitioner | 58 | 4.88±0.46 | 4.78±0.70 | 0.464 |
| | Harm reduction services/Prison Doctor | 9 | 4.11±1.36 | 4.44±0.73 | 0.739 |
| | Opportunistic/Specialist Doctor | 18 | 4.61±0.61 | 4.61±0.70 | 1.000 |
| Recommend Specialist Medical Services | Whole Population | 82 | 4.83±0.49 | 4.83±0.49 | 0.901 |
| | General Practitioner | 54 | 4.87±0.48 | 4.87±0.48 | 0.886 |
| | Harm reduction services/Prison Doctor | 8 | 4.63±0.74 | 4.75±0.46 | 0.564 |
| | Opportunistic/Specialist Doctor | 17 | 4.76±0.44 | 4.71±0.59 | 0.739 |

prescribed in-depth test by the referring doctor after the first suspicion (62.3%; harm reduction services/prison doctor: 83.3%, p=0.013; GPs: 85.3%, p<0.0005). Moreover, the specialist doctor group had a significantly higher proportion of patients who were directly referred to specialised HCV treatment facilities at the time of diagnosis (80%), compared to the GP group (61.6%, p=0.008).

*Care Pathway in Specialist Medicine* – The specialist doctor group had a more positive experience with access to care, reporting easier access to specialist care (p=0.034) and a higher likelihood of finding their appointment locations convenient (p=0.048) compared to the harm reduction services/prison doctor group.

*PREMs* – As shown in Supplementary Table 3 in S1 File, no significant differences in PREMs at T0 were found between the three groups. However, differences emerged at T1 in terms of experiences of the care pathway (Supplementary Table 5 in S1 File). Specifically, patients in the specialist group would recommend primary care services (mean score: 4.48±0.97) less than those in the GP group (mean score: 4.76±0.70, p=0.020). For each group, the time effect of PREMs is illustrated in the relative tables (Tables 3–6).

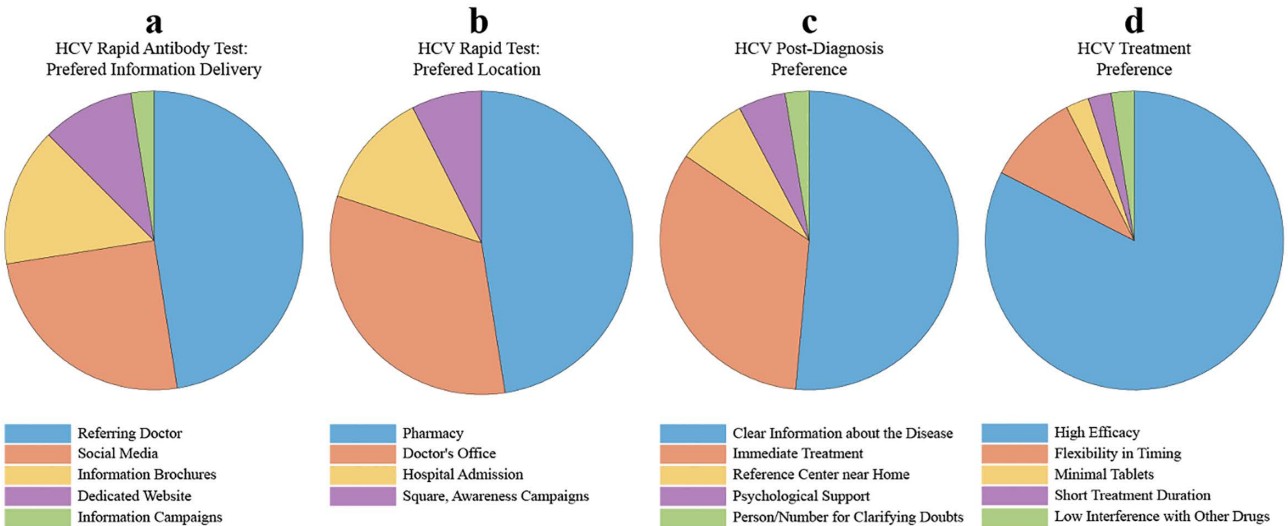

**Fig 1. Pie charts showing the HCV patients' preferences.** Data refer to the topics investigated: HCV rapid antibody tests information delivery (a, n = 40), test location (b, n = 40), post-diagnosis (c, n = 39) and treatment (d, n = 40).

*PROMs* – Supplementary Table 4 in S1 File reveals significant differences in T0 PROMs among the three groups. The harm reduction services/prison doctor group, compared to the GP group, had a higher proportion of patients reporting poorer work/daily performance (p = 0.014) and concentration (p = 0.028) due to emotional state in the past four weeks, affecting social activities (p = 0.001). Additionally, both the harm reduction services/prison and specialist doctor groups reported lower levels of calmness and serenity compared to the GP group. At T0, significant differences were observed in PSC12 (F(2,293)=3.237, p = 0.041) and MSC12 (F(2,293)=7.874, p < 0.0005) among the three groups. Patients who were diagnosed with HCV through a GP exhibited significantly higher MSC12 (44.57 ± 7.08) and PSC12 (48.59 ± 10.79) scores compared to those diagnosed through harm reduction services/prison doctors (MSC12: 39.83 ± 9.42, p < 0.0005) and opportunistic/specialist doctors (PSC12: 44.71 ± 11.2, p = 0.046), respectively.

Moreover, in the whole population, a significant Time effect was observed only in PSC12 (F(2,144)=4.737, p = 0.010), where T0 values (47.71 ± 11.05) were significantly lower than T2 (51.60 ± 10.64, p = 0.014). When considering only HCV patients with known care pathway entry, a significant Time effects for PSC12 were observed in patients arriving at HCV diagnosis thought GP (F(2,94)=3.352, p = 0.039) and thought opportunistic/specialistic doctor (F(2,34)=3.520, p = 0.041). Additionally, a significant Time effect for MSC12 was observed only for patients arriving at HCV diagnosis through opportunistic visit/specialistic doctor (F(2,34)=6.544, p = 0.010). Post-doc analysis revealed a significant improvement between T0 and T1 in PSC12 (T0: 48 ± 11.16; T1: 51.65 ± 8.52; p = 0.025) and MSC12 (T0: 41.56 ± 8.51, T1: 48.06 ± 3.32. p = 0.007) in the groups of general and opportunistic/specialistic doctors, respectively.

Furthermore, the specialist doctor group, compared to the GP group, had a higher prevalence of patients reporting work or daily limitations in the past four weeks (p = 0.024) and higher feelings of discouragement and sadness (p = 0.030). For each group, the time effect of PROMs is illustrated in the relative tables (Tables 5 and 7).

## Discussion

This study reports the outcomes of a regional pilot implementing longitudinal patient-reported experiences and outcomes among individuals with HCV in Tuscany. To our knowledge, this initiative established the first Italian observatory of patient-reported measures in HCV, providing a systematic approach to collecting patient feedback and treatment results. Our analysis yielded several important observations.

**Table 7. MSC12: Time x Group effect decomposition. Paired t-tests were utilised to evaluate differences across time points, with significance levels adjusted via Bonferroni correction.**

|  | Group | n | Time | | | | | |
|---|---|---|---|---|---|---|---|---|
|  |  |  | T0 | T0 vs T1 (p) | T1 | T1 vs T2 (p) | T2 | T2 vs T3 (p) |
| MSC12 | General Practitioner | 48 | 45.10 ± 6.37 | 1.000 | 45.10 ± 7.88 | 1.000 | 45.21 ± 6.37 | 1.000 |
|  | Harm reduction services/Prison Doctor | 5 | 41.80 ± 11.43 | 0.178 | 48.80 ± 9.52 | 0.986 | 44.40 ± 6.69 | 1.000 |
|  | Opportunistic/ Specialistic Doctor | 18 | 41.56 ± 8.51 | **0.007** | 48.06 ± 3.32 | 1.000 | 47.06 ± 4.92 | 0.111 |

Overall treatment outcomes were consistent with findings from other Italian cohorts [36–38] and international data [39,40], confirming the high efficacy of direct-acting antiviral (DAA) regimens in routine clinical practices. Despite the LTFU rate being higher than other European recent data [41], it could be considered a good achievement with respect to other general population courts [42,43]. A comparison of patients who did and did not complete the baseline (T0) questionnaire revealed that younger patients and those with fewer comorbidities were more prone to non-participation and to being LTFU. This observation aligns with other studies [44] and suggests that younger individuals may require targeted outreach strategies—potentially involving digital or community-based support—to increase their engagement in care and completion of patient-reported outcome/experience measures. Few studies deal with this theme [45], and further research is needed to develop strategies to engage young HCV+ patients.

Our findings indicate that patients who enter the diagnostic-therapeutic pathway through specialist consultations receive treatment earlier than those referred by general practitioners or harm reduction services. Such findings highlight an urgent need to optimise HCV screening and referral processes in non-specialist settings, ensuring all patients benefit from timely therapy initiation.

Stratification by referral revealed substantial heterogeneity in patient profiles and needs. Harm reduction clients and people in prison were younger, predominantly male, and characterised by higher psychosocial vulnerability, including psychiatric comorbidities and high-risk behaviours. These characteristics suggest the need for dedicated linkage-to-care strategies that address mental health, social vulnerabilities, and substance-use disorders. These findings are in line with international literature [20,27]. In contrast, GP-referred patients tended to be older, with higher rates of retirement and multimorbidity, but generally reported relatively straightforward access to routine care. This cohort may benefit from closer coordination between primary and specialist care to manage multiple chronic conditions. Patients referred directly by specialists experienced more rapid linkage to treatment but reported less favourable perceptions of primary care services, suggesting potential gaps in continuity and communication between providers. Notably, despite these differences, most patients across all referral pathways reported positive experiences with specialist care (e.g., ease of scheduling visits, short waiting times, accessible locations). Nonetheless, the data suggest that a "one-size-fits-all" approach may not be adequate to address the unique barriers faced by certain high-risk groups, underscoring the need for tailored interventions.

In the overall cohort, patient experiences with referring doctors showed significant improvement from T0 to T1 (e.g., feeling listened to, quality of communication), and these improvements were generally maintained at T2. This pattern suggests a stable perception of care experiences over time and indicates that structural or procedural elements of the healthcare system may not have undergone substantial modifications between these time points. For specialist doctors, PREMs were consistently high at T0 and showed slight fluctuation over time, indicating stable patient satisfaction once individuals were engaged in specialised HCV care. As Jamieson Gilmore et al. reported, these results highlight how PREMs can be key elements in constructing indicators at different health system levels [46].

When analysing PREMs at T0, no significant baseline differences were found between the three patient groups, suggesting that, before treatment, expectations and initial experiences with the healthcare system were broadly comparable across referral pathways. It cannot be ruled out that the questionnaire we administered is not sensitive enough to capture the barriers that people in a situation of vulnerability may encounter. Experiences such as the one documented by Olding et al.[47] suggest how necessary it is to involve the target population of people who use drugs in the construction of the intervention. In contrast, at T1, significant differences emerged in the experiences related to the care pathway, particularly in terms of patients' likelihood to recommend primary care services. Specifically, patients in the specialist group expressed a lower likelihood of recommending primary care services than those in the general practitioner group. This disparity suggests that while specialists provide advanced medical care, their approach may not align as effectively with patients' expectations of primary care services, potentially due to differences in communication, accessibility, or perceived continuity of care.

The results from the analysis of PROMs highlight significant improvements in perceived health status among HCV patients over time. Specifically, patients exhibited a notable enhancement in their perceived physical and psychological well-being from T0 to T1, which remained stable in the long term. This rapid enhancement in health status following DAA therapy is consistent with prior work showing that viral clearance often correlates with a better sense of physical well-being [22,25,26]. The observed trends suggest that access to treatment and follow-up care was crucial in improving health perceptions, reinforcing the importance of structured care pathways. Baseline assessments indicated lower emotional well-being among harm reduction/prison and specialist doctor groups, likely reflecting higher social and medical vulnerabilities within these populations. Although their emotional and social limitations improved over time, the findings highlight the importance of supplementing DAA therapy with mental health resources and social inclusion, especially for individuals with histories of substance use or incarceration [48].

The impact of care pathways on longitudinal PROMs outcomes further supports these findings. While a significant time effect for PSC12 was observed in the entire study population, it was particularly evident in patients diagnosed through general and opportunistic/specialist doctors. This suggests that access to care through general practitioners or opportunistic visits may facilitate a smoother integration into the healthcare system, leading to better long-term outcomes. In contrast, MSC12 improvements were mainly observed in patients diagnosed through opportunistic or specialist doctors, indicating that patients receiving specialised medical attention may experience delayed but significant psychological benefits.

Taken together, PROMs and PREMs provide a nuanced understanding of how patients experience and benefit from HCV care beyond clinical outcomes. They highlight the potential of patient-reported data to identify equity gaps, support performance evaluation, and inform tailored interventions for high-risk groups. Embedding these measures into routine practice may enhance service responsiveness and contribute to progress toward HCV elimination targets.

## Limitations of the study

Despite the valuable insights gained, this study has several limitations. First, although seven out of ten regional prescribing centres participated, the sample may not fully capture all HCV patient experiences across Tuscany, potentially limiting the generalizability of our findings. Second, because clinical data and questionnaire responses could not be directly linked for individual patients, the two datasets had to be analysed separately, reducing the potential for more in-depth correlation analyses. Third, as a retrospective study, our data may be subject to recall bias—particularly for questions related to the timing of diagnosis and patient-reported experiences. A further limitation concerns the progressive reduction in the number of respondents across T0, T1, and T2, which may introduce attrition bias and restrict comparisons across follow-up periods. Several factors likely contributed to this decline. The questionnaire was relatively long and demanding, which may have reduced acceptability and willingness to complete it repeatedly; future work should aim to refine the tool to increase user-friendliness and completion rates. In addition, the modalities of administration may not have been fully optimised for this specific patient population, highlighting the need for methodological improvements to enhance engagement and

adherence over time. Finally, the pilot nature of this work means further research with larger, more representative cohorts is needed to validate and extend these findings.

## Conclusions

In conclusion, this pilot study demonstrates the feasibility and utility of systematically collecting patient-reported data alongside clinical information to evaluate HCV care pathways in Tuscany. Our findings confirm the high efficacy of DAA therapy in achieving SVR12 rates exceeding 93%, with younger patients being more prone to treatment disengagement. Notably, entry into specialised HCV centres is associated with a shorter time from diagnosis to treatment initiation, underscoring the value of direct specialist involvement. Nonetheless, substantial differences exist among referral pathways—particularly for individuals receiving care through general practitioners or harm reduction services/prison settings—highlighting the need for more tailored approaches to screening, referral, and support.

Importantly, our patient-reported outcome and experience measures reveal meaningful improvements in perceived health and satisfaction with care over time, suggesting that viral clearance often coincides with enhanced quality of life. However, the greater prevalence of psychosocial challenges among certain subgroups points to the need for comprehensive programs integrating mental health and social services. Overall, this integrated approach to data collection and analysis provides a foundation for more targeted HCV interventions in high-risk populations and emphasises the importance of collaboration between primary care, specialist providers, and addiction or correctional services to optimise patient outcomes.

## Supporting information

**S1 File. Supplementary tables.** Sociodemographic characteristics, clinical history, referral pathways, patient-reported experience measures (PREMs), and patient-reported outcome measures (PROMs) of HCV patients, stratified by referral source (general practitioners, Harm Reduction/prison services, opportunistic/specialist doctors) and time point. (DOCX)

## Acknowledgments

The authors would like to thank all the members of OPT-HepaC Consortium for their invaluable contributions to the study; Dr. Piero Colombatto; Dr. Lidia Surace; Dr. Antonio Salvati; Dr. Alessia Calì (Hepatology Unit, Pisa University Hospital, Pisa, Italy); Prof. Maurizia Brunetto (Hepatology Unit, Pisa University Hospital, Pisa, Italy; Department of Clinical and Experimental Medicine, University of Pisa, Pisa, Italy); Prof. Cristina Stasi (Link University, Rome, Italy); Dr. Barbara Rossetti; Dr. Cesira Nencioni; Dr. Silvia Chigiotti; Dr. Giulia Ottaviano (AUSL Toscana Sud Est, Grosseto Hospital, Grosseto, Italy); Dr. Danilo Tacconi; Dr. Claudia Bianco; Dr. David Redi (Division of Infectious Diseases, Arezzo Hospital, Arezzo, Italy); Prof. Massimiliano Fabbiani (Department of Medical Biotechnologies, University of Siena, Siena, Italy); Dr. Francesca Panza (Infectious and Tropical Diseases Unit, Siena University Hospital, Siena, Italy); Dr. Sauro Luchi; Dr. Sara Modica; Dr. Sara Moneta; Dr. Sarah Iacopini (Division of Infectious Diseases and Hepatology, San Luca Hospital, AUSL Toscana Nord Ovest, Lucca, Italy); Prof. Anna Linda Zignego (Department of Experimental and Clinical Medicine, University of Florence, Florence, Italy); Dr. Pierluigi Blanc; Dr. Piera Pierotti; Dr. Elisa Mariabelli (Division of Infectious Diseases 1–2, AUSL Toscana Centro, Florence, Italy); and the Management and Healthcare Laboratory of the Sant'Anna School of Advanced Studies for facilitating access to the administrative data.

## Author contributions

**Conceptualization:** Giaele Moretti, Ilaria Corazza, Erica De Vita, Milena Vainieri, Lara Tavoschi.

**Data curation:** Giaele Moretti.

**Formal analysis:** Maria Paola Tramonti Fantozzi.

**Funding acquisition:** Lara Tavoschi.

**Investigation:** Giaele Moretti, Erica De Vita, Milena Vainieri, Lara Tavoschi.

**Methodology:** Giaele Moretti, Maria Paola Tramonti Fantozzi, Ilaria Corazza, Erica De Vita, Milena Vainieri, Lara Tavoschi.

**Project administration:** Lara Tavoschi.

**Resources:** Ilaria Corazza.

**Software:** Ilaria Corazza.

**Supervision:** Erica De Vita, Milena Vainieri, Lara Tavoschi.

**Validation:** Erica De Vita, Milena Vainieri, Lara Tavoschi.

**Visualization:** Maria Paola Tramonti Fantozzi.

**Writing – original draft:** Giaele Moretti, Maria Paola Tramonti Fantozzi, Erica De Vita.

**Writing – review & editing:** Erica De Vita, Milena Vainieri, Lara Tavoschi.

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
