## [Decision Letter · Decision Letter 0]

27 Oct 2025

Dear Dr. De Vita,

We look forward to receiving your revised manuscript.

Kind regards,

Sona Frankova

Academic Editor

PLOS ONE

Journal Requirements:

“his work has been developed within the framework of the project “Optimisation of diagnosis and care Pathways for chronic HCV in Tuscany”, funded by Regional Health Service in the context of the 2018 Health Research Tender (PI: Lara Tavoschi).”

4. In the online submission form, you indicated that “The data that support the findings of this study are available on request from the corresponding author. The data are not publicly available due to privacy or ethical restrictions.”

6. One of the noted authors is a group or consortium “on behalf of OPT-HepaC project team”. In addition to naming the author group, please list the individual authors and affiliations within this group in the acknowledgments section of your manuscript. Please also indicate clearly a lead author for this group along with a contact email address.

Additional Editor Comments:

Methods:

How was fibrosis measured? Please update.

Table 1. Consider providing (in the text or in the table itself) the proportion of PWUD in decreasing-size cohort of 347 – 237 - 104 at T0-to-T2.

The discussion section  is too long and needs re-edition for readability: one gets lost very soon. Your findings are of interest so please give them some juice here.

Whenever possible, include in parentheses numbers and, where appropriate, also p-values. E.g., “younger“ “ among those who completed T0 questionnaire ...“, “commonly reported“, etc.

Limitations paragraph – move from last place to the forefront the issue of progressive decrease in the patient numbers at T0 as concerns questionnaires and also over time.

Reviewer's Responses to Questions

**Comments to the Author**

1. Is the manuscript technically sound, and do the data support the conclusions?

Reviewer #1: Yes

Reviewer #2: Yes

2. Has the statistical analysis been performed appropriately and rigorously?

Reviewer #1: Yes

Reviewer #2: Yes

3. Have the authors made all data underlying the findings in their manuscript fully available?

Reviewer #1: Yes

Reviewer #2: Yes

4. Is the manuscript presented in an intelligible fashion and written in standard English?

Reviewer #1: Yes

Reviewer #2: Yes

Reviewer #1: In their longitudinal prospective study De Vita et al. analyzed clinical and patient-reported outcomes in HCV care in Tuscany, Italy. The authors observed the feasibility and utility of systematically collecting patient-reported data alongside clinical information to evaluate HCV care pathways in Tuscany, and their findings confrim high efficacy of DAA therapy in achieving SVR 12. Their data also confirmed the need for highly individualized care for specific subgroups of HCV patients based on their social situation and differences in patients' treatment experiences depending on the doctor's specialization.

*In the manuscript, the authors reported an SVR 12 rate of 98.6 % when excluding patients lost to follow-up. However, they did not specify whether the achievement of SVR 12 varied according to different stages of liver fibrosis.

* The authors described the individual limitations of the study very well, I have no further comments.

Reviewer #2: Manuscript is well written, statistical analysis is well made, statistical results are clearly presented. Major limitation of this interesting study is a small number of patients who completed questionairres. Discussion is too long and not always targeted to the topic of the research.

**Do you want your identity to be public for this peer review?** For information about this choice, including consent withdrawal, please see our Privacy Policy

Reviewer #1: **Yes:** Mikolas Holinka

Reviewer #2: No

---

## [Author Response · Author response to Decision Letter 1]

31 Dec 2025

Note: For a clearer version,

please see the attached file (rebuttal letter plos).

Dear editor,

Dear reviewers,

Thank you for your time and work. Below, we provided point-by-point answer to the revisions received.

Dear sir,

thank you to select me to review a manuscript: Clinical outcomes and patient-reported measures in HCV care: insight from a longitudinal prospective study in a large Italian region; written by Moretti et al. Authors evaluated clinical and patient-reported outcomes in the HCV care cascade in Tuscany (Italy), offering insights into how health service organization affects effectiveness, equity, and patient experience. 963 patients with chonic hepatitis C from 7 hepatology or infectology centers were treated with DAA; the mean age was 56.31±15.57 years, 61.9% were males. 17.5% had a history of detention, 2.6% were current detainees. 93.4% of patients achieved SVR. The authors evaluated patient-reported outcome measures (PROMs) and patient-reported experience measures (PREMs). PROMs and PREMs were collected at baseline (T0), 3 months post-treatment (T1), and 6 months after T1 (T2). The number of patients that filled the questionnaires were 347, 237, and 104 at T0, T1 and T2, respectively. Patients which filled the T0 questionnaire were older and had less frequent loss of follow-up.

Relative to referring doctor, a significant improvement was observed in all PREMs from T0 to T1. These improvements were generally maintained at T2, except for feelings of being attentively cared by medical doctors, which showed a slight decrease. No significant temporal differences were observed in the PREMs relative to the specialist doctor. HCV patients showed a significant improvement in perceived health status from T0 to T1 (p=0.001), which remained stable over time. Significant differences were observed between T0 and T2: limitations in work/daily concentration and performance due to emotional state or health status.

The specialist doctor group had a significantly higher proportion of patients who were directly referred to specialized HCV treatment facilities at the time of diagnosis (80%), compared to the GP group (61.6%, p=0.008).

SERD/prison services were younger and mostly male. They had higher psychiatric comorbidities and risk behaviors. Patients identified by SERD/prison doctors were notably younger (mean age: 45.89 ± 11.19 years), with higher prevalence of males (90.9%), unemployed (47.1%) and more likely to engage in high-risk behaviors, such as unprotected sex (96.1%) and injection drug use (90.6%). Psychiatric disorders were significantly more prevalent, affecting 33.3% of patients.

Manuscript is well written, abstract and introduction are adequate. Statistical analysis is well made, results are clearly presented. Major limitation of this interesting study is a small number of patients who completed questionairres. Discussion is too long and not always targeted to the topic of the research.

I recommend some changes to improve the quality of this nice manuscript:

1) Please, explain in detail a small number of patients in T0, T1 and T2 group. Please explain the very low number of patients in group T2 and the big decrease in respondents between groups T1 and T2.

We acknowledge that the progressive reduction in the number of respondents across T0, T1, and T2 represents one of the major critical issues of this study. Several factors may have contributed to this decline. First, the questionnaire we developed is relatively long and articulated, which may have reduced its acceptability and the willingness of patients to complete it at multiple time points. Further research is needed to refine the tool, improve user-friendliness, and ultimately increase completion rates.

Second, the modalities of administration were likely not fully optimized for this specific patient population. This aspect also requires careful reconsideration and methodological improvement in future studies to ensure higher engagement and adherence over time.

We’ve provided now adequate justification in the text, in the Limitations of the study paragraph, at the end of the discussion

2) Please specify which DAA regimen patients were treated with.

According to the suggestion of the referee, we added the patients’treatment to the text in the results section. Lines 200-202 (in track-changes version) are now as follow: "Overall, HCV patients were mainly treated with glecaprevir/pibrentasvir (50%) and sofosbuvir/velpatasvir (47.9%), followed by elbasvir/grazoprevir (1.3%) and sofosbuvir/velpatasvir/voxilaprevir (0.8%)." Moreover, these data were added to Table 2, for the entire population as well as for the subgroups who filled and did not fill the T0 questionnaire.

3) Please replace the term echography with abdominal ultrasound.

Thank you, we replaced it.

4) Please specify, how the fibrosis stage was diagnosed.

The methods used by the clinicians for diagnosing liver fibrosis are now explicitly detailed in lines 124–126.

5) Please shorten the discussion.

Thank you for you suggestion, we revised the discussion accordingly, removing redundant paragraph and mantaining the more relevant concepts.

6) English language pollishing is recommended.

Thank you, we revised and polished language.

My final decision is minor revision.

Review Comments to the Author

Reviewer #1: In their longitudinal prospective study De Vita et al. analyzed clinical and patient-reported outcomes in HCV care in Tuscany, Italy. The authors observed the feasibility and utility of systematically collecting patient-reported data alongside clinical information to evaluate HCV care pathways in Tuscany, and their findings confrim high efficacy of DAA therapy in achieving SVR 12. Their data also confirmed the need for highly individualized care for specific subgroups of HCV patients based on their social situation and differences in patients' treatment experiences depending on the doctor's specialization.

*In the manuscript, the authors reported an SVR 12 rate of 98.6 % when excluding patients lost to follow-up. However, they did not specify whether the achievement of SVR 12 varied according to different stages of liver fibrosis.

We thank the referee for the suggestion. No significant difference in liver fibrosis was observed between HCV patients who, twelve weeks after the end of treatment, achieved a sustained virologic response (SVR12) and those who did not (non-SVR12). A chi-square analysis confirmed this lack of difference (χ2(3)=1.646, p=0.649). Indeed, the distributions of the liver fibrosis stages were similar between the two groups (SVR12 - F0/F1: 41.7%, F2: 19.4%, F3: 10.7%, F4: 28.2%; non-SVR12 - F0/F1: 33.3%, F2: 11.1%, F3: 22.2%, F4: 33.3%). This observation is added to the main text, result section (lines 207-209, track-change version).

* The authors described the individual limitations of the study very well, I have no further comments.

Reviewer #2: Manuscript is well written, statistical analysis is well made, statistical results are clearly presented. Major limitation of this interesting study is a small number of patients who completed questionairres. Discussion is too long and not always targeted to the topic of the research.

Thank you for you suggestion, we revised the discussion accordingly, removing redundant paragraph and mantaining the more relevant concepts.

For the number of patients who complete the questionnaire, we explained it in the Limitations of the study paragraph, at the end of the discussion.

---

## [Decision Letter · Decision Letter 1]

12 Feb 2026

Clinical outcomes and patient-reported measures in HCV care: insight from a longitudinal prospective study in a large Italian region

PONE-D-25-40531R1

Dear Dr. De Vita,

We’re pleased to inform you that your manuscript has been judged scientifically suitable for publication and will be formally accepted for publication once it meets all outstanding technical requirements.

Kind regards,

Sona Frankova

Academic Editor

PLOS One

Additional Editor Comments (optional):

Reviewers' comments:

Reviewer's Responses to Questions

**Comments to the Author**

Reviewer #1: All comments have been addressed

2. Is the manuscript technically sound, and do the data support the conclusions?

Reviewer #1: Yes

3. Has the statistical analysis been performed appropriately and rigorously?

Reviewer #1: Yes

4. Have the authors made all data underlying the findings in their manuscript fully available?

Reviewer #1: Yes

5. Is the manuscript presented in an intelligible fashion and written in standard English?

Reviewer #1: Yes

Reviewer #1: In their longitudinal prospective study De Vita et al. analyzed clinical and

patient-reported outcomes in HCV care in Tuscany, Italy. The authors observed the

feasibility and utility of systematically collecting patient-reported data alongside clinical

information to evaluate HCV care pathways in Tuscany, and their findings confrim high

efficacy of DAA therapy in achieving SVR 12. Their data also confirmed the need for

highly individualized care for specific subgroups of HCV patients based on their social

situation and differences in patients' treatment experiences depending on the doctor's

specialization.

ü

The authors answered all of my questions very well. I have no further comments.

**Do you want your identity to be public for this peer review?** For information about this choice, including consent withdrawal, please see our Privacy Policy

Reviewer #1: **Yes:** Mikolas Holinka

---

## [Editor Report · Acceptance letter]

PONE-D-25-40531R1

PLOS One

Dear Dr. De Vita,

I'm pleased to inform you that your manuscript has been deemed suitable for publication in PLOS One. Congratulations! Your manuscript is now being handed over to our production team.

Kind regards,

on behalf of

Dr. Sona Frankova

Academic Editor

PLOS One